# Co-Infection and Cancer: Host–Pathogen Interaction between Dendritic Cells and HIV-1, HTLV-1, and Other Oncogenic Viruses

**DOI:** 10.3390/v14092037

**Published:** 2022-09-14

**Authors:** Tania H. Mulherkar, Daniel Joseph Gómez, Grace Sandel, Pooja Jain

**Affiliations:** 1Department of Microbiology and Immunology, Drexel University, College of Medicine, 2900 Queen Lane, Philadelphia, PA 19129, USA; 2Department of Biological Sciences, California State University, 25800 Carlos Bee Blvd, Hayward, CA 94542, USA

**Keywords:** dendritic cells, HIV-1, HTLV-1, hepatitis viruses, EBV, vaccines, therapeutics, oncogenic viruses, infection and cancer

## Abstract

Dendritic cells (DCs) function as a link between innate and adaptive immune responses. Retroviruses HIV-1 and HTLV-1 modulate DCs to their advantage and utilize them to propagate infection. Coinfection of HTLV-1 and HIV-1 has implications for cancer malignancies. Both viruses initially infect DCs and propagate the infection to CD4^+^ T cells through cell-to-cell transmission using mechanisms including the formation of virologic synapses, viral biofilms, and conduits. These retroviruses are both neurotrophic with neurovirulence determinants. The neuropathogenesis of HIV-1 and HTLV-1 results in neurodegenerative diseases such as HIV-associated neurocognitive disorders (HAND) and HTLV-1-associated myelopathy/tropical spastic paraparesis (HAM/TSP). Infected DCs are known to traffic to the brain (CNS) and periphery (PNS, lymphatics) to induce neurodegeneration in HAND and HAM/TSP patients. Elevated levels of neuroinflammation have been correlated with cognitive decline and impairment of motor control performance. Current vaccinations and therapeutics for HIV-1 and HTLV-1 are assessed and can be applied to patients with HIV-1-associated cancers and adult T cell leukemia/lymphoma (ATL). These diseases caused by co-infections can result in both neurodegeneration and cancer. There are associations with cancer malignancies and HIV-1 and HTLV-1 as well as other human oncogenic viruses (EBV, HBV, HCV, HDV, and HPV). This review contains current knowledge on DC sensing of HIV-1 and HTLV-1 including DC-SIGN, Tat, Tax, and current viral therapies. An overview of DC interaction with oncogenic viruses including EBV, Hepatitis viruses, and HPV is also provided. Vaccines and therapeutics targeting host–pathogen interactions can provide a solution to co-infections, neurodegeneration, and cancer.

## 1. Background

The immune system consists of two parts: innate and adaptive immunity. The innate immune system consists of physical barriers of epithelial cells, mucus secretions lining surfaces in direct contact with external pathogens, constitutively present or secreted soluble proteins and bioactive small molecules that regulate cell function, and membrane-bound receptors and cytoplasmic proteins that bind patterns expressed by invading pathogens [1]. Upon initial exposure to an antigen, the innate immune cells recognize pathogen-associated molecular patterns (PAMPs) via pattern recognition receptors (PRRs) [2]. Recognition of a pathogen signals a specific concoction of cells including macrophages and neutrophils to the site of antigen infiltration depending on the classification of the pathogen. Soon after, the adaptive immune system is activated, allowing for the proliferation of antigen-specific B and T cells [1]. The adaptive immune system creates cells and antibodies that provide the host with lasting immunity. As a result, the next time the same antigen is encountered, the body launches a much quicker and more robust immune response to the threat. Dendritic cells (DCs) function as a link between the innate and adaptive immune responses [3].

DCs are the most potent type of antigen-presenting cells (APC), expressing both self and foreign antigens on major histocompatibility complex (MHC) class I and II molecules [4]. Immature DCs reside in peripheral tissues, scavenging for any foreign pathogens. Upon recognition of an antigen, DCs endocytose the antigens via macropinocytosis and phagocytosis. These antigens are then either processed cytoplasmically or through the endosomal–lysosomal system and are subsequently loaded to respective MHC receptors. MHC class I receptors present intracellular antigens (i.e., viral antigens) while MHC class II receptors present extracellular antigens (i.e., bacterial, protozoa, allergens) [5,6]. These pathogenic antigens are presented to B and T cells within secondary lymphoid organs to trigger a full, robust adaptive immune response. The full immune response involves the proliferation of the antigen-specific B and T cells to create both antibodies that bind and neutralize the pathogen, as well as cytotoxic cells that can effectively recognize infected target cells and induce apoptosis to contain the spread of the virus. DCs migrate from peripheral tissues to secondary lymphoid organs including lymph nodes, the spleen, and tonsils. Migration is accomplished through the upregulation of chemokine receptors such as CCR7 [7]. Within lymphoid organs, the DCs present antigens on MHC receptors to naive T cells, stimulating precise differentiation into cytotoxic T cells (CD8+ T cells) and helper T cells (CD4+ T cells), further divided into TH1, TH2, TH17, and regulatory T cells (Tregs) types [8]. In this way, DCs can act as the key to activating the proper adaptive immune response by regulating the downstream immune response. CD8+ T cells, also known as cytotoxic T lymphocytes (CTLs), are key in the defense against intracellular infections such as viruses and can directly induce apoptosis of target cells [2]. TH1 cells secrete proinflammatory cytokines such as interferon gamma (IFN-γ) and tumor necrosis factor β (TNF-β) to aid in response against intracellular infections caused by viruses and some bacteria, as well as eliminating cancerous cells [8]. TH2 cells secrete interleukin (IL)-4, -5, -10, and -13 which target parasitic organisms, upregulate antibody production, and contribute to allergic response. TH17 cells secrete IL-6, -17, -22 and TNF-α to assist in the activation of neutrophils to fight extracellular bacteria. Under normal homeostatic conditions, in the absence of threat, DCs encounter self-antigens. To prevent a host response to self-antigens and thus autoimmunity, DCs must mitigate the immune response in these conditions. DCs presenting self-antigens stimulate the differentiation of naive T cells into Tregs that function to suppress immunity and inhibit inflammatory response against self-antigens [9].

Within bone marrow, DCs originate from hematopoietic precursor cells of the common myeloid progenitor (CMP) or common lymphoid progenitor (CLP) cell lines located [10]. Accordingly, the major subsets of DCs include myeloid/conventional DC1 (cDC1), myeloid/conventional DC2 (cDC2), and plasmacytoid DC (pDC) [11]. There are many versions of abbreviations for DCs. For simplicity, this paper will focus on and refer to plasmacytoid DCs as pDCs and myeloid DCs as mDCs. Growth factors necessary for DC development and differentiation include the granulocyte-macrophage colony-stimulating factor (GM-CSF) and macrophage colony-stimulating factor (M-CSF) [10]. DCs are located all over the human body and are specialized for each organ they reside in, each type of DC expressing a unique set of CD cell markers. DCs reside in lymph nodes, the spleen, the thymus, blood, the skin, where they are referred to as Langerhans cells, and the gut, where they are especially concentrated in Peyer’s patches. Along with differing locations and CD markers, different types of DCs also have distinctive functions (Table 1). pDCs are primary pro-inflammatory type I IFNs (IFN-α) producers in response to bacterial and viral infections [12]. IFN-α attracts inflammatory cells to the site of the imposing threat. pDCs are also activators of mDCs, helper T cells, and CTLs. mDCs are responsible for antigen presentation and the mediation of T cell responses [13,14]. Noting the differentiation of the DC subtype function is vital in understanding the differential effects of viral infection on each type of DC.

Retroviruses such as human immunodeficiency virus type 1 (HIV-1) and human T cell leukemia virus type 1 (HTLV-1) have managed to evade the immune response by utilizing DCs to their advantage to assist with the infection of T cells. DCs are fundamental in HIV-1 and HTLV-1 retroviral infection and spread. HIV-1 and HTLV-1 initially infect mDCs as an intermediate and eventually spread infection to target CD4^+^ T cells. Both HIV and HTLV are thought to facilitate the propagation of the virus through cell-to-cell transmission via virological synapses to reach their target cell, CD4^+^ T cells [15]. Further clarity on the immune response to HIV-1 and HTLV-1 infection is needed. HIV-1 and HTLV-1 essentially hijack the host immune response and utilize mDCs to their advantage to ultimately reach their target cell. HIV-1 and HTLV-1 initially infect mDCs as an intermediate and eventually spread infection to target CD4^+^ T cells via mechanisms including the formation of virological synapses, viral biofilms, and conduits. HIV-1 and HTLV-1 are both life-long infections that affect millions and can potentially lead to the development of detrimental diseases. Thus, expanding research to determine biochemical pathways involved in retroviral infection of DCs as well as the spread from DCs to target cells is crucial.

In addition to retroviruses, DCs are also thought to be involved in sensing oncogenic viruses belonging to other families. This paper also explores the role DCs play in priming the immune system to combat viruses including Epstein–Barr virus (EBV), hepatitis B, D, and C virus (HBV, HDV, HCV), and human papillomavirus (HPV). EBV, HBV, HCV, and HPV belong to the herpesvirus, hepadnavirus, flavivirus, and papillomavirus families, respectively. EBV, HBV, HDV, HCV, and HPV are all capable of contributing to detrimental viral-induced diseases and oncogenesis. As in retroviral infection, DCs are an emerging focus of research as they are involved in initial sensing and response to primary viral infection. Further research of the DC role in oncogenic viral infection will provide a better understanding of viral infection, elucidating key proteins and checkpoints of infection that could be targeted by antiviral therapies. This article provides a discussion of what is currently known about DC function in retroviral HIV-1 and HTLV-1 infection as well as oncogenic viral infection.

## 2. HIV-1 and HTLV Coinfection

HIV-1, formerly named HTLV-3, was taxonomically separated from HTLVs. HIV-1 and HTLV infection can occur concurrently in coinfection. HIV-1/HTLV coinfection may alter the natural history of general and neurological diseases through different mechanisms [16]. Because HTLV-1 infects CD4^+^ T cells and HTLV-2 infects CD8^+^ T cells, HIV-1 disease progression could vary depending on the HTLV subtype. HTLV-1 virions and proteins upregulate HIV-1 infection by activating CD4^+^ T cells [17,18,19,20]. Peripheral blood mononuclear cells (PBMC) cultures with HIV-1 and HTLV-1 can activate each other [21]. There is also an increase in antigen production and elevation of concentrations of HTLV-1 mRNA expression following coinfection with HIV-1 [22]. Interestingly, the Tax protein of HTLV-1 stimulates HIV-1 replication by activation of HIV-1 LTR [23]. The result of HIV-1/HTLV-1 coinfection leads to the excessive production of defective lymphocytes, catering to the environment’s high levels of HIV-1 virion production [24]. In fact, HIV-1/HTLV-2 coinfection modulates the cellular microenvironment that favors HTLV-2 viability while inhibiting HIV-1 progression [25]. HTLV-2 regulatory protein Tax exerts a protective role against HIV-1 infection. This Tax protein may promote anti-viral immune responses by activation of the canonical NF-kB pathway [26]. HTLV-2 does reduce the activation of T and NK cells and upregulates viral suppressive CCL3L1 chemokine expression. There is an antisense protein of HTLV-2, APH-2, which helps maintain latency by manipulating the transcription [27]. APH-2 could be regulating HIV-1 replication. For example, APH-2 overexpression inhibited HIV-1 Tat to transactivate the HIV-1 LTR-driven expression of luciferase. There is a known motif of APH-2 that downregulates the Tat-mediated transactivation. APH-2 also hampers the virus release by affecting cellular gag expression. Antiretroviral therapy (ART) is successful in controlling HIV disease but has little effect on HTLV-1/2 genome expression; thus, coinfected individuals on ART have an increased prevalence of HTLV-1-associated neurological disease due to minimal impact of therapy on HTLV-1 expression [28].

## 3. Dendritic Cells and HIV-1

HIV-1 affects approximately 40 million people of the world population and has taken over 30 million lives since the start of the pandemic in the early 1980s [29]. With such a large, affected population, it is necessary to study the mechanisms of HIV-1, to further predict potential treatment options. HIV-1 is transmitted through bodily fluids and blood, and can be contracted through sharing needles and sexual intercourse [30]. HIV-1 primarily infects CD4^+^ immune cells [31]. There are multiple stages of infection: acute, chronic, and acquired immunodeficiency syndrome (AIDS), each with characteristic symptoms. Acute infection is characterized by a large initial loss of CD4^+^ cells with an eventual drop of CD4^+^ counts to 200 cells/mm^3^, raising concern for AIDS. The acute loss of adaptive immunity and latency of the HIV-1 virus make HIV a devastating and unpredictable disease. Sequelae of HIV infection include significant weight loss, chronic weakness, increased susceptibility to coinfection by other pathogens due to immunosuppression, neurocognitive decline, and increased risk of developing cancers such as Kaposi sarcoma (KS). HIV-associated neurocognitive disorder (HAND) is characterized by chronic inflammation, neurocognitive decline, and visuomotor and perceptual alterations. HIV is known to enter the central nervous system (CNS) and infect glial cells [32]. HTLV-1 is also believed to invade the CNS, yet the mechanism of how remains uncertain [33].

### 3.1. Immunopathogenesis

Though DCs are less susceptible to HIV-1 infection than CD4^+^ T cells, they are among the first cells to encounter the virus after the infection across the mucosa and play a focal role in establishing HIV-1 infection and progression of the disease [34,35]. DCs are not only able to sense HIV-1 through both CD4 and DC-SIGN, but are also able to be infected by HIV-1 to transmit the virus to CD4^+^ T cells, exploiting the immune response for its own advantage. This section provides an overview of HIV-1 immunopathogenesis highlighting entry and transmission, immune response to the virus, and HIV-induced pathogenesis.

### 3.2. Entry and Transmission

HIV-1 primarily infects CD4^+^ T cells and cells of the macrophage lineage but also infects DCs, B cells, NK cells, epithelial cells, eosinophils, immature bone marrow and thymic precursor cells, Langerhans cells, megakaryocytes, and CNS cells including astrocytes and oligodendrocytes [36]. Existing research heavily emphasizes the role of macrophages and CD4^+^ T cells in HIV-1 infections. Macrophages are among one of the first cell types to be infected by HIV-1 upon primary infection, acting as reservoirs for infection and intermediate cells that transmit the infection to CD4^+^ T cells [37]. HIV-1 env protein binds to host receptors CD4 and DC-SIGN and its homolog L-SIGN [38,39]. In early infection, HIV-1 binds to the CD4 receptor and CCR5 coreceptor on macrophage cell surfaces [40]. HIV-1 binds to the CD4 receptor and CXCR4 coreceptor on T cells at later stages of infection [41].

There is an emerging focus on the role that DCs play in HIV-1 infection as they are one of the first cell types to encounter HIV-1 and they contribute to the pathology of HIV-1 by assisting in viral dissemination [35,42]. HIV-1 utilizes mDCs for infection progression by two general pathways, cis- and trans-infection (Figure 1) [43]. During cis-infection, HIV-1 binds to the CD4 receptor and coreceptor on a DC, as in macrophages and CD4 T cells. The virus is internalized and undergoes viral replication. Trans-infection is completely replication-independent and relies on the host receptor DC-SIGN [44]. When HIV-1 binds to DC-SIGN, the pathogen is internalized, processed, and presented to CD4^+^ T cells on host MHC receptors [43]. Once a DC has been infected or has processed HIV-1 antigens, such as macrophages, it is able to transmit the infection to CD4^+^ T cells through mechanisms such as the formation of virological synapses [45]. Virological synapse formation induced by HIV-1 resembles immunological synapses [46]. Immunological synapses involve the formation of supramolecular activation complexes (SMACs), the center (cSMAC) of which contain T cells interacting with APCs surrounded by a ring of LFA-1/ICAM-1 interaction called peripheral SMACs (pSMACs). The distal SMAC (dSMAC) surrounds the layer of the pSMAC and is composed of dynamic F-actin. In HIV-1 infection, viral gp120 expressing APCs drive virological synapse formation to coalesce in a formation that resembles cSMAC. The formation of the virological synapse creates a controlled, contained environment to allow for the effective spread of HIV-1 from gp120 presenting DCs to T cells.

### 3.3. Immune Response

Innate and adaptive immune responses are initiated to stop the spread of HIV-1 infection. In the early response to infection, HIV-1 PAMPs engage immune receptors TLR2, TLR4, TLR7, TLR8, and TLR9 and triggers cells of the innate immune system such as DCs, monocytes, macrophages, and NK cells generally secrete proinflammatory cytokines including TNF-α, IFN-α, IFN-γ, IP-10, IL-15, IL-18, and IL-22 in a cytokine storm [47,48]. The initial intense inflammatory response is thought to fuel viral replication as HIV primarily targets immune cells. HIV-1 has also evolved mechanisms to resist innate immunity through the downregulation of NK cell activity and complement receptors, PAMPs modification, and RNA-mediated rapid mutations [49]. HIV-1 Nef protein induces endocytosis or retention of MHC class I molecules, decreasing the MHC class I surface expression [50]. Nef protein may play a role in overcoming CD8^+^ T cell and NK cell-mediated recognition through the selective downregulation of TCR ligands HLA-A and -B molecules [51,52].

As part of the innate immune response, DCs sense HIV-1 infection and are either directly infected by HIV-1 or break down the viral components to present to cells of the adaptive immune response. DC-SIGN on the DC surface acts as a PRR for the mannose patterns of the gp120 site on HIV-1 [43]. Whether an adaptive response is initiated is dependent on the structure of N-glycans in the HIV-1 pathogen [53]. The higher the mannose composition of N-glycans of HIV-1, the more efficient binding of HIV-1 to DC-SIGN for adaptative immune purposes [54]. The DC cell is trafficked to various parts of the body via chemokines and their receptors [55]. These APCs move toward the lymph nodes, including those of the CNS, as well as gut-associated lymphoid tissue (GALT) [3,56]. In the lymph nodes, exposed DCs mature and present antigen peptides on MHC molecules, activating the adaptive immune response [57]. B and T cells are activated to produce antibodies and target infected cells, respectively. The antibody is detectable within the serum after 3–4 weeks of HIV infection. Unfortunately, the adaptive immune response is not effective in curing the infection and is considered too little too late. Furthermore, HIV-1 targets CD4^+^ T cells, resulting in CD4^+^ apoptosis and depletion. The markers of CD4^+^ T cell apoptosis including TRAIL, CCR5 microparticles, TNFR2, and soluble FAS ligand increase with HIV-1 viremia [58]. The depletion of CD4^+^ T cells results in an immunocompromised state and increased susceptibility to disease progression of other co-infections, oncogenesis, and viral-associated neuropathogenesis.

### 3.4. Cancer in HIV-1

People living with HIV (PLWH) are at an increased risk of developing cancer regardless of successful ART. HIV proteins such as gp120, negative factor Nef, matrix protein p17, transactivator of transcription Tat, and reverse transcriptase RT induce oxidative stress and can be released from the infected or expressing cells, which is thought to contribute to oncogenesis [59]. Surrounding epithelial cells undergo malignant transformation, and these HIV proteins can act with or without viral oncoproteins originating from viruses such as HCV, HBV, and HPV, which cause the bulk of malignancies of PLWH on ART. HIV-1/HBV and HIV-1/HCV co-infected patients have decreased rates of liver-mediated viral clearance with HIV-1 infection; this accelerates fibrogenesis and increases the rates of liver-related morbidity and mortality, including HCC [60,61]. High HBV and HCV viral loads are critical factors for the progression of liver cancer [62]. Overexpression of viral oncoproteins induces oxidative stress and chromosomal instability and genomic damage, and promotes chronic inflammation, leading to liver damage and malignant transformation of liver cells [63,64].

Brain cancer is highly prevalent in PLWH which includes primary central nervous system lymphomas (PCNSL) and glioblastomas (GBM), though the nature of the brain tumor–HIV-1 relationship is not clearly understood [65,66]. GBM tumors can occur nearly three years after HIV-1 infection [65]. HIV-1 infects several brain cell types which affect astrocytes that serve as a potential reservoir for productive infection, viral persistence, and latency [67,68]. Oncogenesis correlates with angiogenesis factors such as VEGF and VEGFRs, which modulate and support tumor growth [69]. Glioma cells can link with HIV-1 envelope protein gp120. This interaction can increase glycolysis and participate in the Warburg effect that is habitual in malignancy [70]. Both Tat and gp120 induce EMT and cell migration via the TGF-β1 and MAPK signaling pathways [71]. Accessory protein negative factor (Nef) inhibits the apoptotic function of p53 which affects its half-life and DNA binding activity and transcriptional activation [72]. Chronic inflammation is a characteristic of HIV-1 infection and induces a strong oxidative microenvironment. This chronic viral infection results in oxidative stress that allows virally induced cancer to evolve. This drives neoplastic transformation and develops acquired oncogenic mutations in many cells signaling cascades that drive cell growth and proliferation [73]. The deregulation of oxidative stress pathways is correlated with an escalation of ROS production and mitochondrial dysfunction [74]. HIV-associated carcinogenesis can be driven by persistent immune inflammation, dysfunction of B cells, T cells, components of the cells in the innate immune system, and potentially DCs.

### 3.5. Neuropathogenesis

The field of neuropathology places major emphasis on inflammation, aging, and substance use in persons living with HIV (PLWH). The HIV-infected brain pathophysiology has been described in neuroimaging studies that further our knowledge of the neuroinfectious agent. Recent studies comparing levels of neuroinflammation in PLWH and HIV-negative controls report an association between elevated neuroinflammation and poorer cognition in PLWH [75]. Neuroimaging studies beneficially provide insight into structural, functional, and molecular changes made in the brain due to HIV. HIV evokes a strong neuroimmune response and could lead to a cascade of events including chronic inflammation and cognitive decline [76]. PLWH patients continue to have the burden of health complications related to constant HIV infection, such as HAND [77]. HIV enters the CNS soon after initial infection [78]. PLWH has a continued presence of HIV RNA that is detectable in the cerebrospinal fluid (CSF) [79]. Neuroimaging enables noninvasive techniques to repeatedly measure brain changes, and this makes longitudinal assessment simpler. Modern neuroimaging methods can evaluate neuroinflammation involving the injection of tagged radioisotopes using positron emission tomography (PET) and MRI methods, specifically magnetic resonance spectroscopy (MRS). PET imaging has been used to gauge neuroinflammation due to chronic HIV infection. The radiotracer [11C]-PBR28 targets the 19kDa translocator protein (TSPO) of activated microglia. Levels of neuroinflammation are then compared with virologically well-controlled PLWH and HIV-uninfected controls.

PLWH had a significant increase of Myo-inositol within brain regions such as the ventral and dorsal anterior cingulate gyrus, posterior cingulate gyrus, and intraparietal sulcus. Choline was also increased within the dorsal anterior cingulate gyrus of PLWH [76]. PLWH with increasing age has been associated with lower fractional anisotropy (FA), higher mean diffusivity (MD), axial diffusivity (AD), and radial diffusivity (RD), along with multiple white matter tracts relative to age. This was done using diffusion tensor imaging (DTI) and interactions between HIV and aging were the strongest within the fibers projecting to the temporal and frontal lobes [75,80,81]. This is consistent with previous literature implicating frontal–subcortical involvement due to HIV [82]. The visuomotor and perceptual systems observed have alterations in neural activity with joint effects of HIV and aging. There were disruptions in cerebral blood flow, volume, and functional connectivity discovered using functional magnetic resonance imaging (fMRI) in individuals with HIV and frailty [83,84]. The attention network and concentration have been investigated in HIV-positive patients [85]. The blood oxygenation level-dependent functional magnetic resonance imaging (BOLD-fMRI) was performed on 36 subjects (18 HIV and 18 seronegative (SN) controls). They were put through a set of visual attention tasks; the HIV subjects showed similar task performance as SN controls. There was decreased activation in the normal visual attention network (dorsal parietal, bilateral prefrontal, and cerebellar regions), and there was increased activation in adjacent or contralateral brain regions. It is suggested that HIV-associated brain injury leads to reduced efficiency in normal attention networks relying on neural reserves with increased usage to maintain performance during attention-requiring tasks. CCR5 is tethered to G proteins and CXCR4 binds β-arrestin, which could serve as alternative targets for co-receptor therapeutics [86]. β-arrestins can recruit the cellular machinery required for clathrin-mediated internalization, which leads to either receptor being recycled [87,88]. CCR5 and CXCR4 are both G protein-coupled, 7-transmembrane receptors (GPCR). The effectors are MAPKs, GTPases, JAK/STATs, tyrosine kinases, and more, and these include p38 MAPK. Neuroinflammation is carried out by IL-1, IL-6, TNF-α, CCL2, CXCL10, MMPs, and more. This coincides with viral entry and spread, which leads to neurotoxicity and neuronal dysfunction. There is a link found between synaptic compromise, metabolomics of viral reservoirs, and viral persistence with glutamate (Glu). Glu is the most abundant excitatory neurotransmitter in the CNS [89]. In NeuroHIV, the equilibrium of Glu/GABA/Gln is altered and contributes to neuronal and glial dysfunction, as well as cognitive impairment in at least half of the HIV-1-infected population, even in the ART era [90,91,92,93,94,95]. The severity of brain atrophy and dementia can be detected by extracellular levels of Glu in the cerebrospinal fluid (CSF) and plasma [96,97,98]. Neurotoxicity in NeuroHIV pathogenesis is described by molecular mechanistic studies. HIV-1 infection and HAND have an imbalance of the Glu/GABA cycle and an impaired excitatory/inhibitory state of neurons, resulting in the persistent stimulation of NMDARs. A small population of astrocytes become infected and carry HIV-integrated DNA [89]. These astrocytes cannot, however, produce the virus. They contribute to neurotoxicity not blocked by ART which affects neighboring cells (neurons, glia).

### 3.6. Vaccine and Therapeutics

HIV was once considered a fatal disease, but with the implementation of combined antiretroviral therapy (cART), the life expectancy of PLWH is normal in general populations [99]. HIV-1 nanotherapeutics have gone from in vitro to clinical trials [100]. There are some serious side effects and complex dosing regimens with cART; thus, it is limited by several factors. Nanomedicine-based anti-HIV therapeutics (HIV-1 nanotherapeutics) has been an emerging therapeutic technology in recent years, showing success at the preclinical level and/or Phase I/II clinical trials. The four common nanomaterials used in HIV-1 nanotherapeutics include dendrimers, liposomes, micelles, and nanosuspensions that have surfactants, drugs, and phospholipids. There are several advantages of nanoformulated drugs versus free drugs in various applications. Nano drugs overcome anatomical barriers such as crossing the BBB via transcytosis and/or uptake by circulating macrophages. There is active cell targeting of the nano drug to be modified to actively target and deliver drugs to cells through the attachment of receptor targeting ligands. There are benefits of free drugs such as the extension of half-life via nanoformulation. The active ingredient product of nanotechnology is VivaGel (SPL7013 Gel), a leading candidate dendrimer that was specifically designed with human safety in mind [101]. These are microbicides which are compounds that can be applied vaginally or rectally to protect the user from sexually transmitted infections (STIs).

An HIV nanoformulated vaccine has entered Phase II clinical trials and is named the DermaVir patch. It is a promising application of nanomedicine for HIV which is a therapeutic vaccine. The formulation of the patch involves mannosylated polyethyleneimine, glucose, and an HIV antigen-coding DNA plasmid formulated into nanoparticles [100]. The patch delivers nanoparticles to epidermal cells and then engulfs nanoparticles, evolving to produce an immune response. The intention is to use this as an adjuvant to conventional cART. Promising data indicate that the advancement of nanomedicine toward HIV-1 may reduce the healthcare costs of patients. This is critical to delivering these HIV-1 nanotherapeutics in the developing world, where HIV is most prevalent, in a cost-effective manner [102]. The goal of DC vaccines is mounting an HIV-1 antigen-specific T cell response, ideally clearing infection and eliminating the need for long-term ART. DC vaccine trials have been conducted utilizing autologous DCs loaded with HIV-1 antigens. This type of immunotherapy involves loading DCs with antigens ex vivo and then introducing the cells back into the patient [103]. DCs are critical to the recognition of HIV-1, regulation of T cell function, and targeting of infected cells by activation of the adaptive immune system by presenting HIV-1 antigens [104,105]. The use of autologous antigens creates a more “personalized” approach and has the benefit of allowing the extraction of HIV-1 RNA from latently infected CD4 T cells that could not be cleared by ART [106]. During infection, minimizing the emergence of HIV-1 quasispecies is recognized as an important objective of effective DC immunotherapy. The strategies currently being investigated for HIV-1 treatment include inhibiting viral protein function such as ART, Env/gp41 targeting antibodies, CCR5 antagonists, and post-attachment blockers. To reverse latency, there are HDAC inhibitors and PKC agonists. Immune regulation can occur with PD-1 inhibitors, BTLA/CTLA-4 targeting antibodies, and heterodimeric IL-15. Current research focuses on locating areas in the HIV-1 infection pathway that can be used for targeted therapies. For example, this includes inducing inhibitory compounds that may block the CD4^+^ binding site on gp120, preventing pDCs HIV-1 internalization [107]. This approach aims to decrease the spread of HIV-1 infection. Other vaccine approaches attempt to decrease viral load seen in chronically infected patients, for example, by inducing a TH1 response, using inactivated whole virus DC vaccination [108]. Understanding how DCs interact with HIV-1 throughout the body is the first step in teasing out vaccination efforts and treatment options. Imaging is of the spatial distribution of metabolites that are orchestrated by immune events of retroviral lymphoid tissue reservoirs. There has been combined protein and nucleic acid in situ imaging (PANINI) that has multiplex capabilities developed to simultaneously quantify DNA, RNA, and protein levels within tissue compartments. This can also be coupled to multiplexed ion beam imaging (MIBI), in which there are over 30 parameters measured across archival lymphoid tissues. Multiplexed imaging with any platform (MIBI, CODEX, etc.) leads to the identification of cell phenotypes and cellular neighborhoods (CNs). An antibody panel staining with >30 antibodies can be done on structural markers, lymphocytic markers, nuclear factors, macrophage markers, functional markers, and nucleic acids. Using FFPE cell pellets and lymphoid tissues PANINI-MIBI can detect single-integration events of HIV, RNA transcripts, and protein epitopes robustly on the same tissue section. Antibody–oligo-based CODEX platforms and combined PANINI-MIBI capture viral events and tissue morphology [109]. Delineation of lineage-specific markers of diverse immune cells is an application of imaging platforms that can cross-validate antibody markers. The global trends in emerging nanoscale vaccines for infectious diseases include biological, experimental, and logistical problems associated with immune engineering [110]. Vaccine transport and spatial localization in lymph nodes where B-cell receptors (BCR) or surfaces of APCs such as FDCs can flow interstitial fluid localizes in the LN. There are nanobodies that can inhibit CXCR4-mediated signaling and chemotaxis with 238D2 and 238D4 and the data indicates competitive antagonism for both nanobodies.

HIV researchers at the Scripps Institute have investigated a seemingly indestructible HIV-like strain, nicknamed “death star”, for vaccine trials using a harmless laboratory-created adeno-associated virus (AAV) to combat HIV [111]. It is called eCD4-Ig and features two HIV co-receptors, CD4 and CCR5. The viral vaccine is injected into muscle and “infects” muscle cells, which causes them to produce protective eCD4-Ig. HIV is attracted to eCD4-Ig, and thus binds and undergoes conformational change prematurely, and is therefore no longer able to infect. The “death star” research strain is SIVmac239 and is tough to stop. There is currently a Phase I clinical trial investigating three experimental HIV vaccines based on a messenger RNA (mRNA) platform called the HVTN 302 study by the NIAID-funded HIV Vaccine Trials Network (HVTN) expected to be completed by July 2023.

### 3.7. HIV/EBV Coinfection

Most HIV+ individuals also harbor EBV [112]. HIV augments Epstein–Barr virus (EBV)-associated malignancies and is stable or increasing in people living with HIV (PLWH) [113]. EBV is transmitted by saliva, especially in PLWH. The virus may also be shed in semen and vaginal secretions. cART has been shown to be effective with HIV-1 patients and HIV-1-associated neurocognitive disorders (HAND) patients [114], and EBV antiviral agents such as ganciclovir or zidovudine are beneficial but require further investigation [115]. EBV infection is associated with PCNSL which occurs at a higher rate in immunocompromised individuals, which validates that the immune system plays a pivotal role in inhibiting EBV activities [116]. Primarily, EBV latently infects B cells and nasopharyngeal epithelial cells and could interact with follicular dendritic cells (FDCs) in the germinal center (GC). Plasma cells internalize the virus and produce EBV-specific antibodies. It can also infect neurons directly or indirectly through B cell-mediated neuroinflammation and demyelination [117]. In one study, a cohort of participants provided daily oral swabs that were analyzed for the presence and quantity of EBV. The results suggest HIV-1-coinfected individuals have higher rates of B cell reactivation and have higher and more frequent EBV detection in the saliva of an HIV-1-coinfected person [118]. In a recent case study, EBV+ PCNSL in a young (40) female immunocompetent patient, presented with a ring-enhancing lesion in the right basal ganglia. The tumor had surrounding vasogenic edema and promoted a mass effect causing a midline shift and increased intracranial pressure with rapidly progressive neurologic dysfunction [119]. She underwent gross total resection of the tumor with a tubular retractor and after adjuvant therapy with high-dose methotrexate, rituximab, and temozolomide, remains disease-free two years after initial presentation (MRI images of pre-op, post-op, and two years after presentation are provided in the case report). Although this is a rare disease in immunocompetent young individuals, it remains a challenge to evaluate if surgical resection and/or adjuvant chemotherapy are required. On H&E staining, the large tumor cells were characterized by large nuclei, open chromatin, prominent nucleoli, and moderate amounts of cytoplasm. The tumor cells were uniformly positive for CD20. Most tumor cells expressed CD30 and were positive for EBV-encoded small RNAs (EBER) by in situ hybridization. To rule out peripheral involvement, a computed tomography (CT) image of the chest and abdomen can demonstrate enlarged left axillary and subpectoral lymph nodes. It is critical for neurosurgeons and other clinicians to raise awareness of the potential for this rare clinical presentation of EBV+ PCNSL in young immunocompetent patients. It is also imperative for cranial surgeons to evaluate the use of tubular retractor approaches to deep brain tumors if needed. It is important to note that since EBV establishes latency within cells, eradication is impossible.

## 4. Dendritic Cells and HTLV-1

HTLV-1 is the first discovered human retrovirus. HTLV-1 is an enveloped retrovirus with a diploid single-stranded RNA genome in the delta retrovirus group [120]. Although HTLV-1 is present throughout the world, the prevalence and regional estimation remain poorly documented. HTLV-1 is estimated to impact five to ten million individuals worldwide [121]. The key known endemic regions include Japan, sub-Saharan Africa, Central America, South America, the Caribbean islands, focal areas in the Middle East, and Australia–Melanesia [122]. In total, 90% of infected individuals are estimated to be asymptomatic carriers while the remaining individuals develop HTLV-induced diseases such as adult T cell leukemia/lymphoma (ATL), HTLV-associated myelopathy/tropical spastic paraparesis (HAM/TSP), rheumatic syndromes, and HTLV-associated uveitis, among other autoimmune conditions [123]. HAM/TSP is a chronic meningomyelitis of gray and white matter in the spinal cord that presents with spastic paraparesis, impairment of the gait, and autonomic dysfunction of the bowel and bladder.

### 4.1. Immunopathogenesis

HTLV-1 hijacks the host immune system, utilizing the cells to its own benefit. HTLV-1 infects DCs and uses them to propagate infection to its target cells, CD4^+^ T cells. As with HIV-1, DC-SIGN plays a key role in the DC sensing of HTLV-1. Some key proteins for HTLV-1 immunopathogenesis include viral Env and Tax proteins [124]. This section provides a detailed overview of HTLV-1 immunopathogenesis through entry and transmission mechanisms, interaction with the innate and adaptive immune response, and HTLV-induced disease progression.

### 4.2. Entry and Transmission

HTLV-1 is transmitted through cell-containing body fluids and can thus be transmitted through contact with blood products, sexually, and from mother to child through breast milk, [122,125]. HTLV-1 preferentially infects CD4^+^ T cells but has the potential to infect other cell types including DCs, CD8^+^ T cells, B cells, and monocytes. Primary T cells are difficult to infect with HTLV-1 in vitro. They require cell–cell contact and cell-free virus transmission is inefficient [126,127]. DCs, on the contrary, are easily infected with cell-free HTLV-1 and via viral biofilms in vitro. Evidence shows that DCs exhibit a greater degree of binding to HTLV-1 virions than other cell types including T cells [128]. DCs can thus function as intermediates in propagating infection to target T cells via cell-to-cell transmission. DCs facilitate the spread of infection to T cells by trans-infection in which they obtain virions and transfer them to target cells or via cis-infection in which they themselves are infected and then go on to spread the infection to T cells [126].

HTLV-1 envelope glycoprotein (Env) interacts with surface receptors on host cells including DC-specific intercellular adhesion molecule-3 grabbing non-integrin (DC-SIGN), surface glucose transporter 1 (Glut-1), neuropilin-1 (NRP-1), and heparan sulfate proteoglycans (HSPG) to allow viral entry into the target cell, with DC-SIGN being the most important receptor on DCs [129,130]. DC-SIGN facilitates HTLV-1 binding and fusion through an ICAM-dependent mechanism [130].

Cell-to-cell transmission from DCs to T cells depends on specific interactions between the cellular and viral proteins. Two types of cell–cell contacts described are tight cell–cell contacts and cellular conduits (Figure 2) [125]. The tight cell–cell contacts are proposed to involve polarized budding of HTLV-1 into synaptic clefts and cell surface transfer of viral biofilms. Polarized budding at virological synapse requires HTLV-1 Tax protein to enhance the expression of adhesion proteins (ICAM-1) in an HTLV-1-infected cell in contact with an uninfected cell [131]. The engagement of ICAM-1 on the infected cell with LFA-1 on uninfected T cells connects the two cells, creating a synapse between cells. The ICAM-1/LFA-1 interaction stimulates the reorganization of the cytoskeleton in the infected cell, upon which the viral proteins are exocytosed to the virological synapse. The viral proteins concentrate within the virological synapse over time, allowing them to bind and enter the uninfected cell. The virological synapse essentially allows for the directed transmission of HTLV-1 to target cells in a confined space, maximizing the efficiency while minimizing the recognition of viral particles by the host immune response.

Viral biofilms are another mechanism by which infections are spread cell to cell. Studies show that mDCs contain a significantly higher proviral load compared to lymphocytes when exposed to HTLV-1 viral biofilms [127]. Thus, it is more likely for DCs to be infected by HTLV-1 biofilms and spread the virus to T cells via other mechanisms of cell-to-cell transmission, but it is possible that biofilms are still involved. In viral biofilms, HTLV-1 virions and clusters of viral proteins such as Gag and Env are concentrated in a confined, protective environment on the surface of infected cells and are transmitted to target cells at the virological synapses. The viral biofilms are composed of carbohydrates, collagen, agrin, linker proteins such as tetherin, and O-glycosylated surface receptors CD43 and CD45 [132]. The biofilm functions as a reservoir for the viral particles to cultivate before the spread of infection to neighboring cells.

Another mechanism of cell-to-cell transmission involves cellular conduits. Conduits are essentially extensions of the membrane that allow for interactions with cells over greater distances compared to adjacent cell-to-cell contact [133]. It is proposed that HTLV-1 allows for transmission over long distances by the transfer of virions through cellular conduits induced by viral protein p8 [134]. p8 induces LFA-1 clustering on the T cell surface, allowing for T cell adhesion to DCs, conjugating a formation in which the HTLV-1 antigen binds to T cells, and allowing for HTLV-1 transmission [135]. p8 overexpression in chronic HTLV-1 infection promotes polysynapse formation for viral transmission. Overexpression of p8 increases the number and length of the cellular conduits. The exact biochemical mechanism by which p8 promotes HTLV-1 transmission remains unknown.

### 4.3. Immune Response

As with HIV, cell-free HTLV-1 binds and enters mDCs and pDCs peripherally. The DCs then migrate to secondary lymphoid organs where they are exposed to primary T cells to which some DCs transmit the virus through cell-to-cell cis- or trans-infection as above. As this transmission process is occurring, the rest of the host’s immune system also begins launching a combative response against the viral threat from the minute that the virus enters the system. The body begins to fight the virus immediately with an innate immune response. All human host cells express membrane or cytoplasmic receptors that recognize HTLV-1 pathogen-associated patterns (PAMPs) such as HTLV RNA, non-methylated CpG DNA, and envelope glycoproteins [136]. These PAMPs are recognized by pattern recognition receptors (PRRs) such as Toll-like receptors (TLRs) and RIG-I-like helicases (RLHs) expressed by numerous cell types including DCs, macrophages, epithelial cells, and endothelial cells [137]. The PAMP/PRR interaction stimulates IFN regulatory factors IRF3 and IRF7 that promote type I interferon (IFN-I) transcription including IFN-α and IFN-β, which are effective anti-viral mediators [136]. pDCs are the major IFN-I producers. When compared to mDCs, pDCs have an impaired capacity to transfer HTLV-1 to CD4^+^ T cells compared to mDCs, indicating they may be primarily involved in a protective response rather than transmission of the disease to cells [138]. Natural killer (NK) cells are also involved in the innate immune response against HTLV-1, though the exact mechanism of their role is not thoroughly studied. Spontaneous NK cell proliferation positively correlates with the HTLV-1 proviral load [139]. NK cell proliferation, however, is decreased in individuals that develop HAM/TSP and ATL as compared to asymptomatic carriers of HTLV-1 suggesting the impaired function of NK cell proliferation and function in these conditions.

Soon after the initial innate immune response, the adaptive immune response is triggered. In the humoral immune response, B cells form antibodies against HTLV-1 core, envelope, and tax proteins [140]. These antibodies appear in the serum within 30–60 days after the onset of primary infection with anti-gag antibodies predominating. A high HTLV-1 antibody titer correlates significantly with an increased proviral load, indicating that an increased viral load is associated with a more robust immune response. In the cell-mediated immune response, cytotoxic T lymphocytes (CTLs) (mainly CD8^+^) recognize and kill infected cells by interaction with viral antigens presented on MHC class I molecules. Most of the CD8^+^ CTLs recognize and respond to tax protein, indicating a prime target for vaccine therapy. The CD8^+^ CTLs secrete proinflammatory cytokines including IL-2, IL-16, TNF-α, IFN-γ, macrophage-inflammatory protein 1 α (MIP-1α) and 1 β (MIP-1β), and matrix metalloproteinase-9 (MMP-9) [141]. Although CD8^+^ CTL response is protective against most viruses, whether the response is protective or contributory to the inflammatory and demyelinating disease process of HAM/TSP remains under investigation. The CD4^+^ T cells contribute to the adaptive immune response by primarily recognizing env protein [142]. The CD4^+^ T cells consist of effector and regulatory T cells (Tregs). The effector cells assist in the activation of other immune cells and the production of proinflammatory cytokines, while the Tregs modulate the immune response via the expression of transcription factor Foxp3 [143]. Studies suggest that HTLV-1 asymptomatic carriers and healthy controls have higher concentrations of Foxp3+ Tregs versus HAM/TSP patients, supporting that Tregs assist in mitigating the immune response. The immune response against HTLV-1 infection is a very delicate balance as there must be a strong enough response to mitigate the spread of the infection, yet the overstimulation of the immune system also results in detrimental effects towards the host resulting in conditions such as HAM/TSP.

### 4.4. Cancer in HTLV-1

HTLV-1 infection induces ATL in about 2.5–5% of cases [144]. ATL is an aggressive malignancy with a short survival rate that develops more commonly in children infected with HTLV-1 compared to infected adults [145]. The most common symptoms of chronic ATL include a skin rash, swollen lymph nodes, hepatosplenomegaly, and lymphocytosis. ATL can be divided into the aggressive acute, lymphoma, and chronic subtypes as well as the slowly progressing smoldering subtype [146]. The presence of different mutations distinguishes aggressive subtypes from slowly progressing ones such as the IRF4 (interferon regulatory factor 4) or MUM1 (multiple myeloma oncogene 1) gene mutation seen in highly aggressive subtypes. IRF4 is a transcription factor significant in B and T cell maturation believed to have a gain of function mutation and a fivefold increase in expression in aggressive ATL subtypes, making it a potential target for ATL therapies. A higher HTLV-1 proviral load correlates as a risk factor for ATL development [144].

HTLV-1 Tax and HBZ (helix-basic loop zipper protein) are two key proteins involved in ATL oncogenesis [147]. Tax expression is thought to play a role in tumor initiation, primarily through activation of the NF-kB pathway [148,149,150]. The NF-kB pathway promotes lymphocyte proliferation and cell resistance to apoptosis [151]. Tax also interacts with a plethora of other proteins and pathways including NFAT, AP-1, TRAF6, cell cycle inhibitors, cyclins and cyclin-dependent kinases, and PDZ-containing proteins to further promote genetic instability and oncogenesis [151,152,153,154]. Once tumor formation occurs, Tax protein expression is repressed while HBZ continues to be ubiquitously expressed in ATL cells and HTLV-1 cells [155,156]. HBZ promotes the expression of hot genes telomerase reverse transcriptase (hTERT) and JunD, which mitigates the effect of repeated mitosis on cellular senescence, thus promoting the survival of the tumor [157]. Tax and HBZ have opposite effects on several pathways. Tax activates and HBZ inhibits CREB, AP-1, NF-kB, telomerase, and Wnt; HBZ activates and Tax inhibits TGF-β [147]. Tax and HBZ coordinate function in a sophisticated manner to promote oncogenesis.

### 4.5. Neuropathogenesis

Most HTLV-1-infected individuals remain asymptomatic throughout life, but HTLV-1 has the potential to progress to detrimental neuro-inflammatory diseases in humans including HTLV-associated myelopathy/tropical spastic paraparesis (HAM/TSP), rheumatic syndromes, polymyositis, uveitis, Sjogren’s syndrome, and systemic lupus erythematosus (SLE) [123,158]. HTLV-1 induces immune dysregulation, disrupting the balance of cytokines IFN-γ, TNF-α, TGF-β, and IL-10. These key cytokines are responsible for maintaining the balance between inflammatory and anti-inflammatory responses, so this disruption results in the loss of tolerance to host antigens, autoimmunity, and possible chronic inflammation [158]. The mechanisms of HTLV-induced autoimmunity are not completely understood. Molecular mimicry may be involved as well as an increase in pro-inflammatory cytokines IFN-γ and TNF-α with increased T_H_1 response and a corresponding decrease in anti-inflammatory cytokines IL-10 and TGF-β with decreased Treg cell function and decreased tolerance [159].

HAM/TSP is a chronic meningomyelitis of gray and white matter in the spinal cord presenting with progressive spastic paraparesis, impairment of the gait, and autonomic dysfunction of the bowel and bladder. Biopsies of HAM/TSP patients reveal several inflammatory lesions in the brain and spinal cord from active chronic inflammation [160,161]. Magnetic resonance imaging (MRI) of HAM/TSP patients shows the loss of spinal cord volume, suggesting degenerative pathology causing irreversible demyelination with loss of neuronal cell bodies, axons, and glial cells such as astrocytes. The anticipated risk for developing HAM/TSP is about 2% [144]. Risk factors for developing HAM/TSP include the presence of HLA-DRB1*0101 and polymorphisms in genes for pro-inflammatory cytokines TNF-α and IL-15 and stromal cell-derived factor 1 (SDF-1) [162]. Studies in a HAM/TSP Jamaican cohort show decreased pDCs overall, decreased expression of HLA-DR in mDCs, and increased expression of CD86 in pDCs and mDCs [163]. HAM/TSP patients have increased levels of circulating HTLV-1 specific CD8^+^ CTLs which recognize viral epitopes such as tax, rex, and env proteins [164]. DCs exposed to Tax can undergo activation providing constant antigen presentation and co-stimulation to T cells to evoke intense CTL response underlying the condition HAM/TSP [165,166]. Soluble Tax protein binds immature and mature DCs to drive DC activation and maturation to elicit a T_H_1 phenotype. This intense proliferation of HTLV-1 specific T cells potentially cross-react with neural antigens, causing neural dysfunction seen in HAM/TSP. HAM/TSP patients also have higher titers of HTLV-1 specific IgM, IgG, and IgA antibodies both in serum and CSF as compared to asymptomatic carriers [140]. The presence of anti-HTLV-1 IgM antibodies indicates continuous active replication of the virus occurs in this subset of patients.

### 4.6. Vaccine and Therapeutics

HTLV-1 is a lifelong infection without an existing cure; treatment is aimed at symptomatic control and slowing infection progression. The current standard treatment for HTLV-1 post-chemotherapy in Japan is allogeneic hematopoietic stem cell transplantation [167]. This process involves depleting the host’s bone marrow and administrating healthy hematopoietic stem cells to augment the bone marrow function and replace the dysfunctional cells [168]. Preclinical and clinical studies for ART therapy for HTLV-1 based on success in HIV-1 therapy are ongoing [169]. Zidovudine (AZT) is a nucleoside reverse transcriptase inhibitor (NRTI) effective in treating HIV showing a significant decrease in HIV gag and proviral DNA production [170]. AZT was tested against HTLV-1 infection in a rabbit model, showing decreased provirus and lymphocytic infiltration in treated animals compared to untreated controls [171]. AZT trials for HTLV-1 in cell cultures and human subjects showed variable results [169]. Cell cultures exposed to HTLV-1 and AZT simultaneously showed a protective effect on proviral DNA, RNA, and gag expression while exposure to AZT in already infected cells showed minimal effects [172]. Early studies of HAM/TSP patients showed no effect or clinical benefits in AZT monotherapy [173]. The combination therapy of AZT with epigenetic regulator valproic acid (VPA) for treating STLV-1, the simian counterpart of HTLV-1, in baboons showed a 5–12-fold decrease in proviral load in half of the animals tested, showing some promise [174]. A study of AZT and IFN-α combination therapy for ATL treatment resulted in 28% complete remission and 58% generalized major response in the 19 treated individuals, though the mechanism behind why the dual therapy was successful is unclear [175]. Lamivudine (3TC) treatment showed minimal success due to the natural resistance of HTLV-1 to the drug [169]. Other NRTI-focused studies showed that tenofovir (TFV) prodrugs PMPA (9-R-(2-phosphonomethoxypropyl) adenine), tenofovir disoproxil (TDF), and tenofovir alafenamide (TAF) are more effective than TFV against HTLV-1 in cell lines. HIV antiretrovirals are limited in treating HTLV-1 because they are molecularly catered to HIV; thus, further research in refining antiretrovirals for HTLV-1 is needed.

The primary novel HTLV-1 therapy of interest targets Tax protein. One such therapy is a vaccine consisting of autologous DCs pulsed with Tax peptides (Tax-DC) [167,176]. Tax-DC functions to induce HTLV-1-specific CTLs to decrease the proviral load in HTLV-1-infected individuals. In test subjects, Tax-DC maintains the proviral load of HTLV-1-infected patients at a steady level around that of asymptomatic carriers. Other emerging targets for HTLV-1 therapy include DC-SIGN and apolipoprotein B mRNA-editing enzyme catalytic polypeptide-like 3 (APOBEC3) [177,178]. HTLV-1 transmission from DCs to T cells is shown to be primarily mediated by DC-SIGN; blocking of DC-SIGN with anti-DC-SIGN monoclonal antibodies is shown to prevent HTLV-1 transmission from DCs to T cells and decrease the number and size of HTLV-1-induced syncytial formation [130,179]. APOBEC3 enzymes are host-antiviral restriction factors, yet the mechanism is not fully understood. Paradoxically, HTLV-1 tax expression induced increased levels of A3C, A3D, A3F, A3G, and A3H expression in humanized mouse spleen cells [180]. Targeting DC-SIGN and specific APOBEC3 factors could potentially halt viral progression.

### 4.7. HTLV/EBV Coinfection

In an elderly Peruvian HTLV-1 carrier, a diagnosis was made of EBV-positive diffuse large B-cell lymphoma (EBV-DLBCL) [181]. EBV infection induces B-cell lymphotropism and selective immunodeficiency. HTLV-1 induces T cell dysfunction and B-cell proliferation. The combination of EBV and HTLV-1 coinfection and immune-senescence may have played a part in the development of aggressive diffuse B-cell lymphoma. HTLV-1 can potentially activate an EBV promoter [182]. EBV DNA, proviral DNA for HTLV-1, and Tax mRNA were detected in two B-cell lines (Raji with EBV, Jurkat with HTLV-1) from peripheral blood of patients with ATL [183]. In double transfectants (EBV and HTLV-1), there was enhanced expression of adhesion molecules and high levels of interleukin-4 (IL-4) in the selected cell lines and transfectants. This suggests that coinfection activates aggressive organ involvement of enhanced expression of IL-4 signaling, leading to more adhesion molecules. EBV infection is often observed in the clinical course of ATL [184]. ATL-related immunodeficiency might induce EBV-associated DLBCL and associated infection complications [185]. Oncovirus infection (HTLV-1, EBV) triggers EZH1/2 perturbation and H3K27me3 deposition; H3K27me3 reprogramming is a hallmark of cancer and there is no effective therapy developed for H3K27me3-high malignancies harboring EZH2 [186]. There are structures of EZH1/2 dual inhibitors (OR-S1, (R)-OR-S2, and DS-3201, Valemotostat). These inhibitors effectively diminish H3K27me3 and induce gene reactivation.

## 5. DC Interaction with Other Human Oncogenic Viruses

### 5.1. EBV

EBV is a herpesvirus that primarily infects B lymphocytes where the virus establishes a latent infection and can progress to a persistent infection [187]. EBV most commonly causes mononucleosis infection but is also potentially oncogenic with associated development of lymphoid and epithelial tumors including Burkitt’s lymphoma, Hodgkin’s lymphoma, T cell lymphoma, B cell lymphoproliferative syndromes, nasopharyngeal carcinoma, and gastric carcinoma [188]. EBV envelope glycoprotein gp350 binds to CD21 receptors (complement receptor 2) to bind and enter host cells, primarily B lymphocytes along with others such as epithelial cells [189]. Reactivation of EBV results in lytic viral replication predominantly in plasma cells. There are established mechanisms of EBV neuroinvasion and virus-mediated damage in the CNS. EBV can enter the brain by normal B cell trafficking or through infection of brain microvasculature endothelial cells (BMVEC). The endothelial cells of the neurovascular unit (NVU) or infected B cells could be the source of neurotoxicity via inflammatory cytokine release and viral proteins. Specifically, EBV virion (gp350/220) binds to CD21 of a B cell in the periphery and an EBV episome is formed with viral DNA [190]. That B cell crosses the BBB and can interact with EBV-specific T cells. Direct damage occurs by lytic infection of neurons or glia and astrocytes. There is also indirect damage from B cells to uninfected neurons. DCs are thought to sense EBV primary infection and subsequently respond by priming the host’s innate and adaptive immune response via cross-priming [191]. Due to the selectivity of EBV for B and epithelial cells, it is unlikely that EBV infects DCs directly as do HIV-1 and HTLV-1. DCs are proposed to sense and respond to infection through surface TLR3 and retinoic acid-inducible gene 1 (RIG-1) binding to EBV RNA—dsRNA and EBV-encoded small RNAs (EBERs). EBERs are proposed roles in transformation and oncogenesis. EBER expression alone can induce tumors in severe combined immunodeficient mice. EBERs interact with several host proteins to form ribonucleoprotein (RNP) complexes. During the first 24 h of infection, pDCs secrete high levels of type I IFNs to help limit B cell transformation by EBV via recruitment and activation of NK cells [192,193]. DCs activate the NK cells through IL-12, IL-15, and IFN-α [194,195]. The NK cells then go on to produce significant amounts of type II IFNs which help to restrict B cell transformation by EBV for another 3 to 4 days. The NK cells are also able of directly killing EBV-infected B cells undergoing lytic replication [196]. DCs are also presumably involved in priming the adaptive immune response against EBV infection, though their exact role is under debate. While EBV-transformed B cell lines can prime T cells at low frequencies, mDCs are thought to assist with the cross-presentation of EBV antigens from lytic EBV-infected B cells to premature T cells due to increased efficacy of priming with the presence of autologous DCs [197]. Although alternate explanations for the immune response against EBV do exist and clarification is still needed, DCs are proposed to play a central role in activating and priming both the innate and adaptive immune responses.

### 5.2. HBV

Hepatitis B virus (HBV) is a human hepatotropic DNA virus (hepadnavirus) that causes a disease that is mostly spread by exposure to infected body fluids [198,199]. HBV is the primary contributor to liver cancer globally and is ubiquitous in areas of liver cancer [200]. HBV infection is high-risk and can lead to chronic hepatitis, liver cirrhosis, and hepatocellular carcinoma (HCC). Chronic cases need medication and possibly a liver transplant. This serious liver infection is easily preventable by a vaccine. The HBV genome is a circular DNA molecule that is ~3200 base pairs (bp) in length. HBV possesses the smallest genome of any virus known to infect humans. HBV is a member of the family Hepadnaviridae and is a Class VII virus. HBV carries HBV surface antigen (HBsAg) particles. At 17–22 nm in diameter, these particles are filamentous particles, and 45 nm diameter virions consist of an HBsAg envelope and a 36 nm diameter core [201]. The infectious component of HBV is the Dane particle that can be found in HBV antigen-positive sera in patients and through routine electron and immunoelectron microscopy of liver biopsy patients (*n* = 10) [202]. Mature HBV is enveloped, becoming icosahedral virus packages with circular a dsDNA genome with gaps in both strands. It also has a reverse transcriptase covalently bound to the 5′ end of the antisense strand; there are also host proteins included. Within the outer envelope, protein projections are spaced 60 Å apart [203]. HBsAg subviral particles such as spikes have similar spacing yet are arranged in a trigonal geometry on the membrane surface, and there is also an HBsAg lattice that does not align with the core’s icosahedral lattice. The HBV capsid has a dynamic metabolic compartment that participates in structural changes to encapsulate its genome. The interaction of the envelope capsid interaction is unsteady, and the envelope proteins have multiple structures. HBV is the lion’s share of liver cancer globally and is ubiquitous in areas of liver cancer. HCC oncogenesis involves dysregulation in several cellular signaling pathways including Wnt/FZD/β-catenin, PI3K/Akt/mTOR, IRS1/IGF, Ras/Raf/MAPK, and pERK/JAK/STAT [204,205]. HCC screening by liquid biopsy of cell-free DNA (cfDNA) is an early-stage detection of HCC. It has been developed to identify HCC from the surface antigen of hepatitis B virus-positive asymptomatic individuals in the community population [206]. HBV immunization interrupts mother-to-infant transmission and is a preventative measure of cancer. DNA can be extracted from sera and peripheral blood lymphocytes on a commercially available kit (High Pure Viral Nuclei Isolation Kit, Roche Diagnostics, Canada). HBV-DNA can be tested by a real-time polymerase chain reaction (PCR) using the LightCycler Real-Time PCR HBV. The WHO guidelines recommend an abdominal ultrasound and alpha-fetoprotein (AFP) measurement every six months. There are several phases of clinical biomarker validation such as: Phase I: preclinical exploratory to study cancer tissues to identify biomarkers; Phase II: clinical assay validation for cancer cases versus control validation; Phase III: retrospective longitudinal studies for validation in cancer subjects prior to cancer diagnosis; Phase IV: prospective screening for prospective evaluation of stage of tumor detection; and Phase V: cancer control studies to evaluate the impact on the cancer burden in a population. There are limitations to the sensitivity of AFP alone for HCC detection and only 40–50% of HCCs do not have elevated levels of AFP. A secretory antagonist of the Wnt/B-catenin signaling pathway is Dikkopf-1 (DKK1) which is a glycoprotein. The DKK1 concentrations were significantly high in HCC patients compared to the control [207]. DCs play a role in the immunopathogenesis of chronic HBV infection; the DC function of patients with chronic HBV infection is impaired, which may contribute to viral persistence. Laboratory efforts have conducted a recombinant human granulocyte-macrophage colony-stimulating factor (rhGM-CSF) combined with IL-4 which is a classic culture combination to DCs. Combining IFN-λ with rhGM-CSF and rhIL-4 can increase the expression of DC surface molecules and secretion of IL-12 and IFN-γ in patients with chronic HBV infection [61]. This could all lead to a potential DC vaccine that can induce DCs’ maturation to treat chronic hepatitis B. Multikinase inhibitor sorafenib monotherapy is the standard of care for patients and is known to show survival benefits in advanced HCC [208].

### 5.3. HDV

HDV is an unclassified satellite virus that depends on HBV infection to infect target cells. HBV/HDV coinfection happens when a person is simultaneously infected with both HBV and HDV, whereas an HDV superinfection occurs when someone is already chronically infected with HBV (CDC). HDV is known as a “satellite virus” because it can only infect people who are also infected by HBV. The patient coinfected or superinfected with HBV/HDV can result in liver cirrhosis and liver failure. HDV infection is uncommon in the United States, although notably, the actual number of HDV cases in the US is unknown [209]. Significant liver disease is a result of the hepatitis delta virus which is a blood-borne virus that infects hepatocytes. HDV is a single-stranded RNA virus and its propagation and replication are reliant on envelope proteins of HBV to achieve the assembly and release of infectious virus particles [210]. When an individual is exposed to HDV, it is called superinfection. HDV is a small 36 nm single-stranded negative-sense RNA virus with a small RNA of about 1700 nucleotides and one with a circular conformation [211]. HDV genome replication is totally different from HBV. Delta antigens form multimers and HDV RNA wraps around such multimers. Recent experimental situations have shown that HBV envelope proteins are sufficient for the assembly of infectious HDV even in the absence of HBV replication.

Signal transduction of immune mechanisms involving DCs and CD4^+^ T cells and CD8^+^ T cells involves secreting IL-12 and presenting antigen with the MHC-I pathway that activates these T cells. Intrahepatic HDV replication in HBV transgenic mice produced a positive immunohistochemical staining pattern of HDAg in HBV transgenic mice at 7 days after hydrodynamic injection. Dendritic cell-derived exosomes (Dexs) have been discovered to induce immune responses that are competent in eliminating viruses [212]. To induce a robust immune response, biological carriers deliver antigens that are used as immunomodulators from exosomes derived from mature dendritic cells exosomes (mDexs). MDexs are enriched by proinflammatory cytokines secreted by DCs. Designed exosomes are loaded with the ubiquitinated HDV small antigen (Ub-S-HDAg) and then mice are treated, replicating HDV with these exosomes to investigate their antiviral effect and mechanism. There is a need to develop new therapeutic strategies to eradicate HDV in the future. HDV infection increases the risk of hepatocellular carcinoma (HCC) compared to HBV monoinfection. HDV has a ribozyme (HDV-Rz) that is around 67 nucleotides and can be used as an effective tumor gene therapy. HDV-Rz is the fastest of the known naturally occurring self-cleaving RNA and it stands out due to its stability to denaturants. The catalytic activity of HDV-Rz is required for viral replication and viral particle assembly within host cells. HDV-Rz has a self-scission property that cleaves its own sugar–phosphate backbone using a combination of metal ion and general acid–base catalysis through a cytosine nucleobase C75 that acts as a general acid and a divalent metal ion (Mg^2+^) that acts as a Lewis acid [213]. The HDV-Rz has the potential to be used to inhibit hepatocellular carcinoma (HCC). A previous study designed an HDV ribozyme against the RNA component of human telomerase in HCC cell lines, as normal hepatocytes and other cancers, and examined the function of HDV-Rz [200]. The total structure weight is 57.96 kDa and the crystal structure of HDV-Rz has been solved. There have yet to be cryo-EM structures of this ribozyme. There has been a comparison of secondary structures, tertiary folds, and cleavage sites of HDV and hatchet ribozyme products [214]. Gene therapy for cancer treatment focuses on efficacy and safety, leading to the efficient suppression of cancer cells without harming normal cells [215]. Point mutations that occur at a high incidence in cancer can be used as potential targets for specific ribozyme-mediated reversal of the malignant phenotype.

### 5.4. HCV

HCV is a small, enveloped, plus-strand RNA virus belonging to the family Flaviviridae and the genus Hepacivirus. HCV is a liver-trophic virus that chronically infects approximately 170 million people worldwide. The HCV RNA genome encodes a single polyprotein that is post-translationally cleaved into 10 polypeptides including three structural proteins (core, E1, and E2) located at the N-terminus, and five nonstructural proteins. DC-SIGN and L-SIGN are two novel HCV envelope binding receptors, which the HCV envelope glycoprotein E2 binds through high-mannose N-glycans [216]. DC-SIGN and L-SIGN on the membrane are expressed with the highest affinity. The infected hepatocyte detects viral RNA with Toll-like receptor (TLR) 3 or TLR3 and RIG-I, and the infected hepatocyte deattenuates IFN and ISG gene expression. pDCs act as guardians in HCV-infected livers. The ability of an infiltrating pDC to secrete type I IFN typically depends on cellular sensors that can detect the presence of DNA and RNA viruses. pDCs express a subset of specialized cellular sensors named TLR8 and TLR9, which sense endosomal RNA and DNA, respectively. Direct exposure of HCV RNA to pDCs elicits an IFN response in a TLR7-dependent manner. Observations suggest that HCV RNA is transferred from infected hepatocytes to pDCs that trigger IFC response by activation of TLR7. The mechanism of RNA transfer is unknown but could potentially occur through vesicle-mediated transfer by autophagosomes or exosomes [217,218,219]. DC-based vaccines against HCV have been developed. There are some being developed at the experimental level and some toward the translational level. DC-based vaccine against HCV infection includes a DC treated with the fusion protein, with challenge inoculums of anthrax toxin fusion protein containing the HCV core epitope and was done on HCV core-specific CTLs in mice [220]. DC transduced with recombinant adenovirus with challenge inoculums recombinant adenovirus expressing HCV core protein [221]. Another DC transfected with adenovirus encoding NS3 protein from HCV (AdNS3) was used with an outcome of a multi-epitopic CD4 T helper cell 1 (T_H_1) and CD8 T cell responses in different mouse strains [222]. DC transfected with lentiviral vectors (LV) were used, where the LV expressed HCV structural or non-structural gene clusters. There was a potent stimulation of CD4 and CD8 T cell allogenic and autologous responses [223]. DC-containing microparticles of NS5 protein-coated microparticles had an antigen-specific CTL activity in mice and significantly reduced the growth of NS5-expressing tumor cells in vivo [224,225]. There are also DCs pulsed with HCV-LPs where the challenge inoculums were HCV-LP core and HCV-LP E2, and this outcome resulted in HCV core-specific CD4 and CD8 T cell responses [226]. There are strategies to enhance the efforts of DC-based vaccines [227]. Adjuvants include glucopyranosyl lipid A (GLA) which is a new synthetic non-toxic analog of lipopolysaccharide (LPS). Within four hours, GLA caused DC to upregulate CD86 and CD40 and produce cytokines including IL-12p70 in vivo [228]. Synthetic oligodeoxynucleotides (ODNs) that contain unmethylated CpG motifs trigger cells expressing TLR9 (human pDCs, B cells) to mount an innate immune response (T_H_1 production and pro-inflammatory cytokines). CpG ODNs improve the function of APCs and boost the generation of humoral and cellular vaccine-specific immune responses. Clinical trials have demonstrated CpG ODNs to have a good safety profile and increased immunogenicity of co-administered vaccines [229]. High levels of IL-10 in chronic HCV infection have been insinuated for the poor antiviral cellular immune responses. This is an immunosuppressive effect on APCs such as DC. Diaz–Valdes’ group has developed peptide inhibitors of IL-10 to restore DC functions and induce efficient antiviral immune responses. It is a cutting-edge technique to chase metal ions bound to protein to resolve ion interactions within proteins; the zinc-finger A20 is a negative regulator of the TLR and TNF receptor signaling pathways. This was found to play a pivotal role in controlling the maturation, cytokine production, and immunostimulatory potency of DC [230]. A20 has been put through mechanistic studies that tell of regulation in DC production of retinoic acid and pro-inflammatory cytokines. These are some of the prospects of a DC-based vaccine against HCV infection. HCV infection has major implications for cancer and cancer treatment. There is an interplay between HCV-induced oncogenic signaling pathways such as the EGF pathway: HCV binding to its entry receptor complex (CLDN1/CD81) induces EGFR phosphorylation. This is encouraged by the action of NS3/4A and NS5A which negatively regulate the phosphatase TC-PTP and the process of EGFR degradation. There is also the STAT3 pathway where STAT3 activation directs the core protein indirectly via EGFR activation and the NS5A protein that favors the production of ROS. There are also two miRNAs such as miR-135a-5p and miR-19a that decrease the expression of the negative STAT3 regulators PTPRD and SOCS3 [231]. There is also a pro-oncogenic inflammatory microenvironment induced by HCV. Viral sensors such as RIG-I and TLR3 lead to the production of type I IFNs. This occurs during HCV infection in hepatocytes. Macrophages sense the HCV-infected hepatocytes that trigger NLRP3 inflammasomes which induce the secretion of IL-18 which activates NK cells. STAT3 takes part in MDSCs producing IL-10 for the expansion of regulatory T cells. This chronic liver inflammation leads to cell death, hepatocellular regeneration, and the emergence of mutated cells that may be premalignant [232].

### 5.5. HPV

Oncogenic HPVs are recognized to be connected to head and neck squamous cell carcinomas (HNSCCs) that develop from mucosal epithelium in the oral cavity, pharynx, larynx, and sinonasal tract [233]. Cervical adenocarcinoma in in situ, anal, vulvar, and vaginal intraepithelial neoplasia is caused by HPV types 16, 18, 31, 33, 45, 52, and 58. High-risk HPV oncogenes have signal pathways that include two known viral oncoproteins: E6 protein inactivates tumor suppressor p53 mediated DNA damage and apoptotic pathway. E7 protein inactivates the tumor suppressor pRb-mediated cell cycle regulation pathway. Cervical cancer is the fourth most common cancer among women globally, with more than 95% of cervical cancer cases attributable to human papillomavirus (HPV), a double-stranded DNA (dsDNA) virus. Cervical and other anogenital cancers are associated with infection of “high-risk” HPVs: mainly HPV16 and HPV18. Mature DCs are essential in the immune response against HPV [234]. DCs exposed to the high-risk HPV (HR-HPV) induced tumor microenvironment may attenuate effector T cell functions through programmed death receptor 1 (PD-1)/programmed death ligand 1 (PD-1/PD-L1) immune inhibitory signaling [235]. Interactions between DCs and T cells by the ligation of PD-L1/PD-1 and immunoregulatory cytokine secretion may allow for immune escape during chronic HR-HPV infection. PD-1/PD-L1 axis activation causes a loss of function of CTL cells which can be restored by the administration of antibodies that blocked the PD-1/PDL-1 interaction [236]. Cervical cancer cells may also favor tumor establishment by inhibiting the migration of DCs to lymph nodes (via depletion of CCR7) and inducing DC to produce matrix metalloproteinase (MMP) 9 or MMP-9 [237].

Current HPV therapies include monoclonal antibody cetuximab with radiation, immune checkpoint inhibitors pembrolizumab and nivolumab for recurrent or metastatic HNSCC, and HPV vaccines such as Gardasil 9. HPV vaccination of all children and young adults with HPV challenges will dramatically reduce HPV-positive HNSCC. HPV vaccines can prevent vaginal and vulvar cancer, genital warts, anal cancers, mouth, throat, head and neck cancers, and HPV-positive HNSCC in both women and men [238]. Studies reported clinical trials based on DC vaccines for cervical cancer and HNSCC [239]. DCs are very cytotoxic towards HPV16 E6 and E7 proteins expressing cells [240]. Many vaccines include adjuvants that enhance natural immunity, but their mechanisms of action are not fully understood. Adjuvant therapy targets include TLR ligands, CpG oligodeoxynucleotide (CpG ODN), LPS, and polyinosinic–polycytidylic acid (poly(I:C)) a synthetic of double-stranded RNA (dsRNA). These adjuvants aim to pulse DC activity. There is a novel HPV nanovaccine that could effectively inhibit the progression of cervical cancer by combining nanotechnology and photodynamic therapy [241]. The nanovaccine links bovine serum albumin (BSA) with the E7 antigen and then encapsulates the photosensitizer and adjuvant through disulfide bonds to form a highly biocompatible and stable structure. This nanovaccine is a targeted delivery approach to the lymph nodes that could effectively induce the maturation of DCs and stimulate T cell effects, thus enhancing antitumor immunity. This strategy can be applied to the design of a therapeutic HPV vaccine in the future for clinical use.

## 6. Conclusions and Perspectives

DCs are important mediators of innate and adaptive immunity. They are among the first cells to encounter viral infection and thus play an integral role in responding to and propagating infection in several viral infections including HIV-1, HTLV-1, EBV, HBV, HDV, HCV, and HPV. DCs are ubiquitously involved in responding to viral infections. DCs sense HIV-1 and HTLV-1 with surface receptors such as CD4 and CCR5/CXCR4 for HIV-1; Glut-1, NRP-1, and HSPG for HTLV-1; and DC-SIGN for both. mDCs then either transmit the infection to CD4^+^ T cells through cis- or trans-infection via methods including virological synapse, conduit, biofilm, and cSMAC-like structure formation. DCs are also thought to be involved in sensing other oncogenic viruses. EBV is highly selective for infecting B cells and is thus less likely to infect DCs. DCs sense EBV through TLR3 and RIG-1 expressed on the DC surface binding to EBV RNA. DC function in HBV infection is impaired, contributing to viral persistence. Dexs are exosomes that can functionally eliminate HDV infection by targeting HDAg. pDCs are mainly involved in HCV detection through TLR7, 8, and 9 binding to endosomal RNA and DNA. DCs exposed to oncogenic, high-risk HPV diminish T cell functions through PD-1/PD-L1 interactions. HPV-induced cervical cancer cells also inhibit immune responses by inhibiting the migration of DCs to lymph nodes through the decreased expression of CCR7 and increased expression of MMP-9 on the DC cell surface. Overall, DCs have various functions in viral disease progression propagating viral functions to other target cells through increased function in retroviruses versus functioning to mainly suppress infection in most oncogenic viruses. DC-SIGN is an enticing potential target for therapies aimed at stopping viral disease progression at the first step of infection. Cell-to-cell connections are influenced by ICAM-1/LFA-1 surface protein interactions which are influenced by viral proteins such as HIV-1 gp120 expression on MHC II molecules and HTLV-1 Tax protein that plays central roles in viral oncogenesis. Key mediators of HIV- and HTLV-induced neuropathogenesis include Myo-inositol and choline, Glu/GABA imbalance, NMDA receptor signaling, and Foxp3+ Tregs. Further understanding the integrative roles of these factors in viral pathogenesis is essential to developing targeted therapies. Current therapies for HIV-1 and HTLV-1 include cART, HIV-1 nanotherapeutics, and vaccine therapies. cART treatment is effective in HIV-1 treatment and has been tested in HTLV-1 as well. However, these medications including AZT, 3CT, and TFV are designed for HIV-1 and are thus not as effective for HTLV-1. Modification of these medications could optimize the use for HTLV-1. HIV-1 nanotherapeutics are composed primarily of dendrimers, liposomes, micelles, and nanosuspensions that have surfactants, drugs, and phospholipids. The nano drugs are aimed at targeting delivery to specific cells to contain the viral spread. The current standard HTLV-1 therapy is hematopoietic stem cell transplantation. Novel HTLV-1 therapies are centered around Tax protein with the Tax-DC vaccine being the main one. Other potential targets for HTLV-1 therapy include DC-SIGN and APOBEC3. Viral-induced oncogenesis contributes to a significant proportion of cancer cases and can be caused by viruses in several different families as discussed. Viruses promote the development of a variety of cancers such as Burkitt’s lymphoma, Hodgkin’s lymphoma, T cell lymphoma, nasopharyngeal carcinoma, gastric carcinoma, Kaposi sarcoma, hepatocellular carcinoma, and cervical adenoma. Among the oncogenic viruses, DCs are important cells that intercept the virus early in infection. Overall, DCs provide a promising focus for research in retroviral infection and other oncogenic viruses.

## Figures and Tables

**Figure 1 viruses-14-02037-f001:**
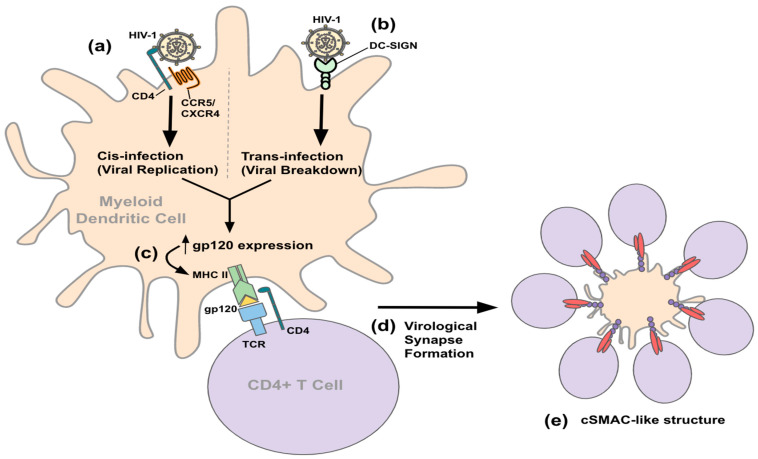
HIV-1 interaction with mDC. (**a**) HIV-1 binds the CD4 receptor and corresponding coreceptor (CCR5/CXCR4) on the DC cell surface, inducing internalization of the virus, viral replication, and cis-infection where the DC itself becomes infected with HIV. (**b**) HIV-1 binds the DC-SIGN receptor on the DC cell surface leading to viral breakdown and trans-infection where the DC merely processes HIV-1 and presents particles to other immune cells. (**c**) Both cis- and trans-infection lead to increased HIV-1 gp120 protein expression on MHC class II surface receptors. The DC MHC II receptor presenting gp120 binds to TCR and CD4 receptors on the T cell surface. (**d**) Increased gp120 expression induces virological synapse formation that resembles the cSMAC structure that is seen in immunologic synapses. (**e**) The cSMAC-like structure is composed of a DC that is surrounded by a ring of CD4^+^ T cells. The cells are connected via ICAM-1/LFA-1 interactions where ICAM-1 is expressed on the DC and LFA-1 is expressed on the T cell.

**Figure 2 viruses-14-02037-f002:**
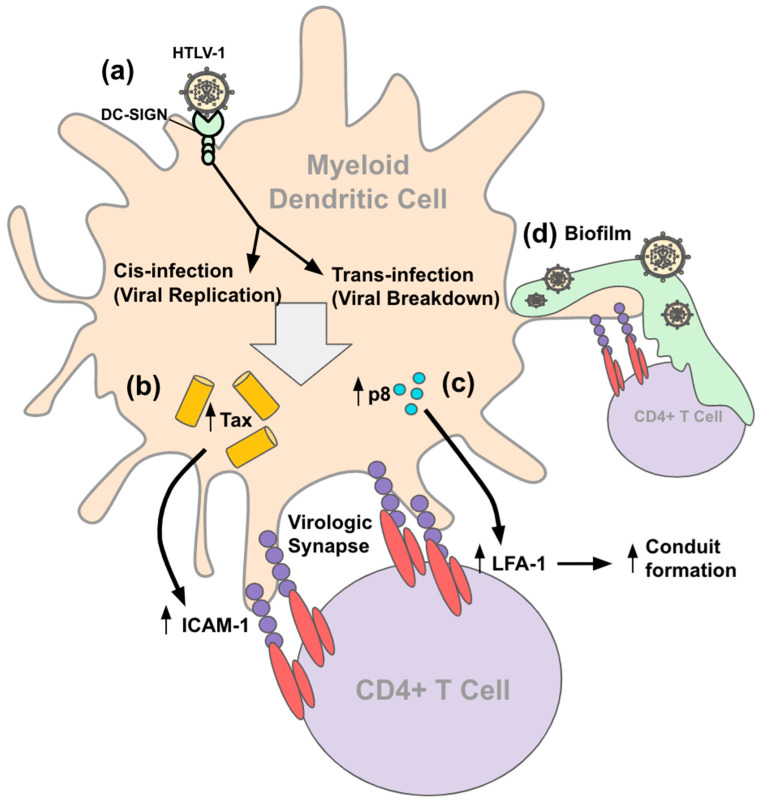
HTLV-1 interaction with DC. (**a**) HTLV-1 binds DC-SIGN, Glut-1, NRP-1, and/or HSPG receptor on DC cell surface. DC-SIGN is the most important receptor on DCs. HTLV-1 then enters the DC and undergoes viral replication to infect the DC in a process called cis-infection. Alternatively, HTLV-1 enters the DC and undergoes viral breakdown to be processed and presented on MHC surface receptors (not pictured) in a process called trans-infection. DCs spread HTLV-1 infection to CD4^+^ T cells through the formation of virological synapses, conduit formation, and viral biofilms. (**b**) HTLV-1 Tax protein accumulates in the DC and induces increased ICAM-1 expression on the surface of the DCs. Increased ICAM-1 allows for more efficient binding to CD4^+^ T cells via LFA-1 interaction. ICAM-1/LFA-1 interaction allows for the formation of virological synapses where HTLV-1 components are exocytosed to and concentrated within to increase exposure of T cells to viral particles for infection. (**c**) HTLV-1 p8 protein also builds up within the DC and induces increased LFA-1 protein expression on T cells. Increased LFA-1 expression functionally allows for conduit formation between DCs and T cells to allow for the spread of infection over greater distances versus neighboring cells. (**d**) Viral biofilms form on the surface of the DC and act as reservoirs for viral growth to allow the spread to neighboring CD4^+^ T cells.

**Table 1 viruses-14-02037-t001:** CD markers on dendritic cell subtypes. Breakdown of DC populations by location including lymph nodes, spleen, thymus, blood, skin, and gut. Differential CD expression among this population of cells. The table also notes proliferative factors for select DC subtypes.

DC Location	CD Molecule Expression	Cytokines and Signaling Molecules
Lymph Nodes	CD80, CD86, CD40	CCR7, CCL19/CCL21, CXCL9/10,
Spleen	CD8, Esam(hi)CD11b+, CD205, CD207, Clec9a	Proliferate in presence of Flt3L, Notch2
Thymus	CD8, CD8a, CD207, CD1a, CD11c	SIRPα+ dependent on CCR2 chemotaxis, S100,
Blood (pDC, cDC, iDC)	pDC: BDCA2, BDCA4 cDC: CD11	CCR7, CCR2, CXCR3, MHCI, IL-12, IL-2, IL-15, CCL3, CXCL8, TNF
Skin (Langerhans)	CD207, CD45, CD11c, CD1a, CD14, CD141	Proliferate independent of Flt3; Dependent on CSF-1R which induces chemokines CCL2 & CCL20, CCR7, CCL21, CCR2, IFN-λ, TLR3, IFN-α/β, IL-12p40, XCR1, IDO, TLR2/8, CD5, TLR7
Gut	CD8, CD103, CD207, CD11b, CD70	Proliferate in presence of Flt3L, Notch2, ALDH1A1/2, RA, IL-5, IL-6, TGFβ, CX3CR1, TLR5, Flagellin, TSLP, IL-23

## Data Availability

Not applicable.

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
