# Peer review of "Co-Infection and Cancer: Host–Pathogen Interaction between Dendritic Cells and HIV-1, HTLV-1, and Other Oncogenic Viruses"

_viruses, 2022, doi:10.3390/v14092037_

Round 1
Reviewer 1 Report
The present manuscript by Mulherkar et al. reviews a very important and probably underappreciated aspect of host-pathogen interactions, namely the ones occurring between chronic human viral pathogens (HIV-1, HTLV-1, and various oncogenic viruses) and dendritic cells. The authors thoroughly and comprehensively review the published literature on the subject for each viral pathogen. Overall, the manuscript is clearly written and well-structured. However, there are a few points should be addressed to improve the manuscript.
The review contains a significant amount of information that does not appear to be strictly related or necessary to convey the main point of the subject. This is especially true for the sections focusing on HIV-1 and HTLV-1. Some the information not directly or strictly linked to the main subject of the manuscript could be eliminated or mentioned in a more concise manner referencing relevant literature where more details can be obtained. This would help the reader focus on the most critical notions.
In the Background section, it should be noted that HTLV-1 is not the first retrovirus to be discovered. Rous Sarcoma Virus was the first retrovirus discovered, while HTLV-1 was the first human retrovirus to be discovered.
The authors state that several HIV-1 proteins (gp120, Nef, Tat, p17) are oncogenic (page 7). However, they do not cite specific research articles showing the ability of these proteins to promote malignant transformation. In fact, the only literature proposing direct oncogenic potential by an HIV-1 encoded protein are studies on the matrix protein, p17. It is very important to use caution when stating that HIV-1 (or its products) are oncogenic. There is a significant difference between stating that HIV-1 suppresses immune responses and thus aids other factors (KSHV, EBV, HTLV-1, etc.) in causing malignancy and stating that HIV-1 (or its proteins) have oncogenic potential. Therefore, the authors are invited to revise their statement or to provide publication demonstrating the ability of HIV-1 proteins to directly cause malignant transformation.
The section that discusses the impact of HIV-1 infection on brain malignancies (page 8) takes an unexpected and unexplained turn when the authors indicate that HIV-1 may increase cervical carcinogenesis. This sentence appears somewhat out place in this section.
The title of section beginning at line 332 of the manuscript (Cancer malignancies in HIV-1) appears redundant. In addition, several sentences within this same section appear very similar to sentences found in reference #78. It would be advisable to revise sentences or sections that may appear too much alike the ones in reference #78.
A section that draws parallels or contrasts among the human pathogens discussed in this manuscript and their interaction with DC would also be very useful. It would allow the reader to tie together a wealth of information that is available in the literature and that the authors have discussed in their manuscript.
The sentence at lines 413-415 appears incomplete.
The sentence at lines 415-416 should be revised.
Author Response
Response to Reviewer 1 Comments
Point 1: The review contains a significant amount of information that does not appear to be strictly related or necessary to convey the main point of the subject. This is especially true for the sections focusing on HIV-1 and HTLV-1. Some the information not directly or strictly linked to the main subject of the manuscript could be eliminated or mentioned in a more concise manner referencing relevant literature where more details can be obtained. This would help the reader focus on the most critical notions.
Response 1: HIV-1 and HTLV-1 sections edited to make information more organized and concise.
Point 2: In the Background section, it should be noted that HTLV-1 is not the first retrovirus to be discovered. Rous Sarcoma Virus was the first retrovirus discovered, while HTLV-1 was the first human retrovirus to be discovered.
Response 2: Specified that HTLV-1 is the first human retrovirus to be discovered.
Point 3: The authors state that several HIV-1 proteins (gp120, Nef, Tat, p17) are oncogenic (page 7). However, they do not cite specific research articles showing the ability of these proteins to promote malignant transformation. In fact, the only literature proposing direct oncogenic potential by an HIV-1 encoded protein are studies on the matrix protein, p17. It is very important to use caution when stating that HIV-1 (or its products) are oncogenic. There is a significant difference between stating that HIV-1 suppresses immune responses and thus aids other factors (KSHV, EBV, HTLV-1, etc.) in causing malignancy and stating that HIV-1 (or its proteins) have oncogenic potential. Therefore, the authors are invited to revise their statement or to provide publication demonstrating the ability of HIV-1 proteins to directly cause malignant transformation.
Response 3: Statement altered to say these factors directly promote oxidative stress.
Point 4: The section that discusses the impact of HIV-1 infection on brain malignancies (page 8) takes an unexpected and unexplained turn when the authors indicate that HIV-1 may increase cervical carcinogenesis. This sentence appears somewhat out place in this section.
Response 4: This unexpected discussion was removed from the paper.
Point 5: The title of section beginning at line 332 of the manuscript (Cancer malignancies in HIV-1) appears redundant. In addition, several sentences within this same section appear very similar to sentences found in reference #78. It would be advisable to revise sentences or sections that may appear too much alike the ones in reference #78.
Response 5: The Title of the section (Cancer malignancies in HIV-1) functions to separate the discussion of malignancies vs. neuropathogenesis. Sentences resembling source #78 revised.
Point 6: A section that draws parallels or contrasts among the human pathogens discussed in this manuscript and their interaction with DC would also be very useful. It would allow the reader to tie together a wealth of information that is available in the literature and that the authors have discussed in their manuscript.
Response 6: Summary discussion added to the conclusion section.
Point 7: The sentence at lines 413-415 appears incomplete.
Response 7: This sentence was out of place--deleted this sentence.
Point 8: The sentence at lines 415-416 should be revised.
Response 8: Sentence revised.
Reviewer 2 Report
Dear Editor,
This deep review is very helpful for those non-expert in infectious diseases/imunnology field. I would say, however, the basic concepts should be summarized since the readers should have such background for this field
Author Response
Thank you reviewer 1 for your response that states our deep review is very helpful to non-experts in infectious diseases/immunology field. I believe it will serve both non-experts and experts alike. We have made the basic concepts more concise to account for the background readers will have for this field.
Reviewer 3 Report
This review investigates the host-pathogen interactions between dendritic cells, HIV-1, HTLV-1, and oncogenic viruses. The article is well written and well articulated. The review is exhaustingly long and would benefit from just sticking to the topic sometimes. Some of the information on HIV/HTLV seems repetitive in the manuscript. The background should be shortened considerably and literature about each viruses should be moved to their respective sections from the background.
Comments
1.The references to geographical regions are irrelevant for the topic of the MS.
2.Line 161-164 does not belong/make any sense.
3.there does not seem to be clear flow/order to which the authors state therapies and signaling pathways often. See line 168
4. sentences like this comes out of nowhere. HPV-positive HNSCC is on the rise [29]. Line 185.
5. Lines 383-387 -irrelevant.
6.Line 443-455-irrelevant.
Author Response
Thank you for your words of encouragement such as being well written and well articulated. I understand that this is a long review as this is a topic that needs to be covered through to this audience. HIV-HTLV coinfection is the topic of the special issue. We have divided the sections into parts that start with the coinfection (presence of both viruses) and then separated the two viruses in sections regarding interactions with DCs. We did shorten the background considerably and move the literature about each virus into their respective sections. The references to geographical regions have been removed. Life 161-164 was removed. I addressed the flow and order of therapies and signaling pathways. I have reworded Line 185. Lines 383-387 were considered and re-stated. Prof. Dr. Jain has rearranged and reworded the section that included Line 443-455. Thanks again for your thorough review!
Round 2
Reviewer 1 Report
The authors have done an excellent job in revising their first manuscript, and they have addressed all the weaknesses that were highlighted in the first round of review.
The only minor issue remaining is the one pertaining to the title of section discussing malignancies in HIV-1 infection. The reason we suggested revising the title is simply that the words “cancer” and “malignancy” are essentially synonymous. Therefore, the title “Cancer malignancies in HIV-1” contains two words with similar meaning, and it is redundant. It is recommended to change the title and use only one of the two words.
Everything else looks great.
Author Response
Point: The only minor issue remaining is the one pertaining to the title of section discussing malignancies in HIV-1 infection. The reason we suggested revising the title is simply that the words “cancer” and “malignancy” are essentially synonymous. Therefore, the title “Cancer malignancies in HIV-1” contains two words with similar meaning, and it is redundant. It is recommended to change the title and use only one of the two words.
Response: Thank you for clarifying this point. The section title was altered to "Cancer in HIV-1."